# Elucidating the Role of *SlBBX31* in Plant Growth and Heat-Stress Resistance in Tomato

**DOI:** 10.3390/ijms25179289

**Published:** 2024-08-27

**Authors:** Qiqi Wang, Xiangqiang Zhan

**Affiliations:** State Key Laboratory of Crop Stress Biology for Arid Areas, College of Horticulture, Northwest A&F University, Yangling 712100, China; wangqiqiwqq1989@163.com

**Keywords:** tomato, *SlBBX31*, transcriptome analysis, photomorphogenesis, flowering and fruiting process, plant growth, heat-stress response

## Abstract

Heat stress inhibits plant growth and productivity. Among the main regulators, B-box zinc-finger (BBX) proteins are well-known for their contribution to plant photomorphogenesis and responses to abiotic stress. Our research pinpoints that SlBBX31, a BBX protein harboring a conserved B-box domain, serves as a suppressor of plant growth and heat tolerance in tomato (*Solanum lycopersicum* L.). Overexpressing (OE) *SlBBX31* in tomato exhibited yellowing leaves due to notable reduction in chlorophyll content and net photosynthetic rate (Pn). Furthermore, the pollen viability of OE lines obviously decreased and fruit bearing was delayed. This not only affected the fruit setting rate and the number of plump seeds but also influenced the size of the fruit. These results indicate that *SlBBX31* may be involved in the growth process of tomato, specifically in terms of photosynthesis, flowering, and the fruiting process. Conversely, under heat-stress treatment, *SlBBX31* knockout (KO) plants displayed superior heat tolerance, evidenced by their improved membrane stability, heightened antioxidant enzyme activities, and reduced accumulation of reactive oxygen species (ROS). Further transcriptome analysis between OE lines and KO lines under heat stress revealed the impact of *SlBBX31* on the expression of genes linked to photosynthesis, heat-stress signaling, ROS scavenging, and hormone regulation. These findings underscore the essential role of *SlBBX31* in regulating tomato growth and heat-stress resistance and will provide valuable insights for improving heat-tolerant tomato varieties.

## 1. Introduction

In the ever-changing global climate, the influence of heat stress on plant growth and development has become a top topic in agricultural research [1,2]. Although the adverse effects of heat stress on plants are well documented, the intricate mechanisms underlying these phenomena remain rarely known [3,4]. Previous studies delved into the multifaceted impacts of heat stress on plants which highlighted not only the direct physiological consequences but also the intricate molecular mechanisms involved [3,5]. 

Heat stress has profound and immediate physiological effects on plants. These impacts can range from subtle changes in metabolic processes to disruptions in growth and development [5,6]. Heat stress has been proved to inhibit plant growth, particularly in terms of stem and leaf expansion [7]. In addition, the efficiency of photosynthesis, a key process for plant growth and productivity, is also compromised under heat stress [8,9]. These direct physiological reactions determine the overall health and yield of crops. 

At the molecular level, plants use a range of strategies to cope with heat stress. One such strategy is heat-stress signaling transduction. The core of this signaling lies in the heat shock transcription factor (HSF)—heat shock protein (HSP) pathway [10,11]. Under heat-stress, proteins can misfold, and these inactivated proteins are detected by certain HSPs [12]. Another strategy involves the regulation of hormone levels which have an impact on plant growth and development, such as abscisic acid (ABA), ethylene (ETH), jasmonic acid (JA), salicylic acid (SA), and gibberellins (GA) [13,14,15,16,17,18]. However, the balance of these hormones is disrupted under heat stress, leading to growth inhibition. Furthermore, plants possess a robust antioxidant system that helps mitigate the damage caused by the ROS generated [19]. Finally, when the ROS production exceeds the antioxidant capacity of the plant, oxidative damage occurs, further compromising plant health [20].

In the plant kingdom, B-box domain-containing proteins constitute a subclass of zinc-finger proteins, notable for their one or two B-box domains located within the N-terminal regions. Certain members of this subclass possess a CCT (CONSTANS, CONSTANS-like, and TIMING OF CAB1) domain. Based on the configuration of their B-box and CCT domains, along with the presence of VP (valine–proline) motifs, BBX proteins are organized into five structural groups. Group I members are distinguished by their two consecutive B-box domains, accompanied by one CCT domain and a VP motif. Similar to Group I, Group II proteins also exhibit two B-box domains and a CCT domain, but lack a VP motif. Members of Group III bear a single B-box domain along with a CCT domain. Group IV proteins are marked by the presence of two B-box domains, without a CCT domain. Lastly, Group V comprises proteins that solely feature one B-box domain [21]. The BBX proteins alter various biological processes in plants [21,22,23]. Members of the BBX protein family are known for regulating photomorphogenesis, the developmental process in which plants respond to light [23,24]. They interact with photoreceptors and other transcription factors to modulate gene expression which affects plant growth, development, and adaptation to light environments [25]. BBX proteins participate in the formation of the circadian clock mechanism [8] which modulates plant physiological processes such as photosynthesis, metabolism, and stress responses [26]. BBX proteins are instrumental in plant stress responses. They can regulate the expression of stress-responsive genes and assist plants in coping with environmental stresses such as drought, salinity, and extreme temperature [27,28]. BBX proteins interact with other proteins, such as transcription factors, photoreceptors, and enzymes [22,29,30]. These interactions allow them to participate in complex regulatory networks which include coordinating plant growth, development, and adaptation to environmental changes.

Based on our heat-stress transcriptome data, we were interested in a particular transcription factor gene, *SlBBX31*. Recent studies indicated that *SlBBX31* enhances chilling-stress tolerance [31]. However, its specific effects on tomato growth and heat-stress tolerance remain undocumented. This laid the foundation for our utilization of reverse genetics methods in exploring the function of *SlBBX31*. In this study, overexpressing of *SlBBX31* in tomato resulted in decreased chlorophyll content due to alterations in photosynthesis and chlorophyll biosynthesis pathways. Moreover, we found that knocking out *SlBBX31* improved heat tolerance by reducing the accumulation of ROS and regulating the expression of various *HSF-HSP* genes during heat stress. Overall, our data indicate that *SlBBX31* acts as a negative regulator of plant growth and heat-stress tolerance in tomato.

## 2. Results

### 2.1. Sequence and Phylogenetic Analysis of SlBBX31

In order to investigate the evolutionary relationship of the BBX protein family, we conducted a phylogenetic analysis of 31 members of the tomato BBX protein family and 32 members of *Arabidopsis* (*Arabidopsis thaliana* (L.) Heynh.). *SlBBX31* (*Solyc06g063280.1*) was notable for its predicted open reading frame (ORF) of 786 bp, which can be converted into a protein composed of 261 amino acids. SlBBX31 possessed a molecular mass of 28.72 kDa and a predicted isoelectric point (pI) of 9.14. Evolutionary analysis showed that SlBBX31 protein shared the highest homology with AtBBX30, AtBBX31, and AtBBX32, all of which belong to subfamily V (Figure 1). Similarly, through a comparative analysis of SlBBX31 with homologous proteins from *Arabidopsis*, tobacco (*Nicotiana benthamiana* L.), potato (*Solanum tuberosum* L.), pepper (*Capsicum annuum* L.), and apple (*Malus domestica* Mill.), among others, we discovered a conserved B-box domain at the N-terminus (Appendix A). Tomato SlBBX31 shared the closest phylogenetic relationship with potato StBBX32 (Appendix A).

### 2.2. Expression Patterns of SlBBX31

To ascertain the subcellular localization of the SlBBX31 protein, the vector 35S-SlBBX31-GFP was transiently expressed in tobacco leaves (Figure 2a). The signals from the SlBBX31-GFP fusion protein overlapped with those from the DAPI staining which served as a positive control for nucleus localization. The fluorescence of SlBBX31-GFP was specifically observed in the nucleus, while the fluorescence of the control GFP was dispersed across the entire cell. Because of the fact that the nature of the target protein, host, expression vector system, codon preference, and transformation conditions can affect the expression of heterologous proteins [32], we hypothesized that our low transformation efficiency was due to the characteristics of the vector. Perhaps a more suitable carrier could enhance the transfection efficiency. According to our recent data, SlBBX31 is likely to function in the nucleus.

We conducted quantitative real-time PCR (qRT-PCR) analysis to examine transcript levels of *SlBBX31* in various plant tissues, including roots, shoots, leaves, flowers, and fruits. *SlBBX31* transcripts were detectable across all tissues, with the highest expression levels detected in flowers and the lowest in roots.

### 2.3. SlBBX31 Expression Is Responsive to Heat Stress

In the initial phase of heat stress, spanning from 0 to 12 h, no alterations occurred in the expression level of *SlBBX31*. (Figure 2d). Particularly, this was largely induced after 24 h of heat-stress treatment, and its expression level decreased to the normal level during the recovery period, which was comparable to the level during the untreated state (Appendix A). The qRT-PCR results indicated a similar expression trend of *SlBBX31* to the transcriptome analysis results and the expression peaked at 24 h as well (Figure 2e). The notable induction of the expression pattern of the marker gene *HSP70-1 (Solyc09g011030.3)* [33] observed under heat-stress conditions served as compelling evidence for the credibility of our heat-stress experiment (Figure 2f,g). The above results suggest that *SlBBX31* expression is responsive to heat stress.

### 2.4. Overexpression of SlBBX31 Inhibits Chlorophyll Accumulation and Photosynthesis in Tomato

To further explore the function of *SlBBX31* in tomato, we conducted genetic transformation and screening on T_0_ generation positive seedlings. Through self-crossing and an additional screening processes, we successfully obtained two independent T_2_ generation homozygous lines with overexpressed *SlBBX31*, named OE-2 and OE-5, respectively. At the same time, we obtained two knockout homozygous lines using CRISPR gene-editing methods, labeled as KO-11 and KO-20. In the OE-2 line, the expression level of *SlBBX31* obviously increased, reaching approximately 300 times of the original expression. Similarly, the OE-5 line showed an increase of nearly 150 times of *SlBBX31* expression (Appendix A). The KO-11 line featured a double base deletion, whereas the KO-20 line featured a single base insertion (Appendix A). Both mutations led to premature translation termination, ultimately resulting in short truncated proteins (Appendix A).

In comparison to the wild-type (WT) plants, chlorosis (loss of green color) was observed in the leaves of OE lines nearly one month old (Figure 3a). To find out the cause of this visible chlorosis, we measured the chlorophyll content in the leaves. There was a remarkable increase in chlorophyll a content and apparent decrease in chlorophyll b content in the leaves of three OE lines compared to the WT plants (Figure 3b,c). As a result, there was an evident decrease in total chlorophyll content (Figure 3d). Under field greenhouse conditions, we further measured the parameters related to photosynthesis. The Fv/Fm value represents the maximum photochemical quantum yield of photosystem II (PS II), or the maximum efficiency of light energy conversion in PS II. The Fv’/Fm’ result reflects the effective photochemical quantum yield of PS II and the primary light energy capture efficiency of the PS II reaction center [34]. Compared to the wild-type, both OE lines presented considerable decreases in Fv/Fm (Figure 3e) and Fv’/Fm’ values (Figure 3f), as well as an obvious reduction in net photosynthetic rate (Figure 3g). Nevertheless, there were no clear differences between the two KO lines and the WT. 

A transcriptome approach was employed to illustrate the distinct effects of *SlBBX31*. We focused on identifying differential expressed genes (DEGs) which displayed opposite upregulation and downregulation patterns in the OE and KO lines. Under normal growth conditions (Heat-0 h), we found 112 differentially expressed genes which were downregulated in the OE lines and simultaneously upregulated in the KO lines. Additionally, 72 genes were upregulated in the OE lines and downregulated in the KO lines (Appendix A). GO (Figure 4a) and KEGG (Figure 4b) enrichment analyses revealed that these 184 genes accounted for plant responses to red, blue, and far-red light, perception of light signals, chloroplast development, photosynthesis, and the MAPK signaling pathway. Developmental phenotypes, such as leaf chlorosis and fruit color changes (Figure 5a), which occurred in OE lines under field growth conditions, may be attributed to these DEGs. From the GO enrichment, we chose the genes linked to the most notable chlorophyll binding term. In the same way, based on the KEGG enrichment, we picked the genes associated with the most prominent photosynthetic antenna protein pathway. These genes belong to the light-harvesting complex (LHC), members of the antenna protein family which exhibit analogous expression patterns. Their expression levels were largely downregulated in OE lines while being upregulated in KO lines (Figure 4c and Appendix A). LHC proteins are transmembrane proteins which affect photoprotection mechanisms. They can capture light energy and rapidly transfer it to reaction centers (PS II and PS I) to initiate photochemical reactions [35]. By combining transcriptome data with the phenotypic characteristics of OE lines, we hypothesized that the substantial reduction in *LHC* expression levels of OE lines affected the antenna protein content. These data prove that *SlBBX31* is an inhibitor of chlorophyll accumulation and photosynthesis in tomato.

### 2.5. Overexpression of SlBBX31 Affects the Reproductive Growth Process in Tomato

In this study, fruits of similar size and from the same growth stage were selected for observation. During the mature green (MG) and breaker ripe (BR) stages, the fruits of the OE lines appeared paler in color compared to those of the WT and KO lines (Figure 5a). By cross-sectioning the fruits, it was observed that the mature fruits of the OE lines generally contained a lower number of seeds, and these seeds appeared less plump (Figure 5b). During field cultivation, pollen abortion was noticed in the OE lines. Through pollen viability tests, it was found that the pollen viability of the two OE lines was below 40%. The OE-2 lines had even lower pollen viability compared to the OE-5 lines, possibly due to a higher expression level of *SlBBX31* in the OE-2 lines (Figure 5c). As a result of the negative impact on pollen viability, the OE lines presented a considerable reduction in fruit setting rate (Figure 5d). The OE lines had a massive reduction in the average number of seeds per fruit (Figure 5e). The fruit-bearing branches location of the OE lines were notably higher than that of the WT and KO lines, which indicated a clear delay in fruit-bearing (Figure 5f). The average fruit weight of the OE lines was significantly reduced compared to the WT and KO lines (Figure 5g). Similarly, the average fruit diameter substantially decreased to less than 40 mm (Figure 5h). Nevertheless, no evident disparities in the above indexes were observed between the KO lines and the WT. These results suggest that *SlBBX31* overexpression affects the reproductive growth process in tomato.

### 2.6. SlBBX31 Is Important for Heat Tolerance in Tomato

According to preliminary experiments conducted under controlled environmental conditions, it was detected that the plants showed a yellowing phenotype when they were cultured to the one-month-old seedling stage with a specific nutrient solution and light intensity. After being subjected to heat treatment at 42 °C for 24 h, the entire plants wilted completely, and the growing points had necrotic damage. When the test materials which underwent heat-stress treatment for 24 h were returned to the artificial climate chamber for a 5-day recovery period, the survival rates of the various lines were calculated. The survival rates of OE-2 and OE-5 plants were only 5% and 10%, respectively, while the survival rate of the wild type was 75%. In contrast, the survival rates of the KO-11 and KO-20 lines exceeded 90% (Figure 6b). Based on the above preliminary experiments, as it was not suitable to select the 24-h heat-stress treatment time point for sampling and measuring physiological indicators, the formal experiment designed a heat-stress treatment duration of 12 h.

Subjected to a 12-h heat tolerance test at 42 °C, the OE-2 and OE-5 lines had the most severe leaf wilting, while the KO-11 and KO-20 lines presented minimal wilting, with a majority of the plants remaining strong (Figure 6a). When plants are subjected to adverse environmental stresses, cell membranes are particularly vulnerable. The relative ion permeability (electrolyte leakage) can qualitatively assess the damage degree to cell membranes caused by heat stress [27]. After 12 h of heat-stress treatment, the conductivity of the OE lines was evidently higher than the control, while the conductivity of the KO lines was apparently lower. The OE lines suffered more severe damage to their cell membranes, whereas the KO lines manifested minimal damage, which implied a stronger resistance to heat stress in the KO lines (Figure 6c). These data indicate that *SlBBX31* is critical for heat-stress tolerance in tomato.

### 2.7. SlBBX31 Is a Negative Regulator for ROS Detoxification under Heat Stress

ROS, an umbrella term for oxygen-containing compounds within plants, accumulates in response to heat stress. ROS are vital signaling molecules for plant growth and development. But excessive accumulation can cause potentially severe damage to plant tissue [27]. We employed qualitative and quantitative methods to measure the contents of H_2_O_2_ and O^2•−^. NBT and DAB staining results showed that both H_2_O_2_ and O^2•−^ accumulated at identical levels under normal growth conditions among these lines. After a 12-h heat-stress treatment, the OE lines had deeper staining compared to the WT, while the KO lines exhibited lighter staining (Figure 6d). Assessments of H_2_O_2_ and O^2•−^ were quantified to verify the histochemical staining results. H_2_O_2_ and O^2•−^ were found to accumulate obviously more in the OE lines as the heat treatment progressed (Figure 6e,f). 

Under adverse environmental conditions, plants can enhance the activity of antioxidant enzymes such as SOD, POD, CAT, and APX to eliminate the excessive accumulation of ROS, and thus to minimize cellular damage caused by endogenous ROS [20]. After heat-stress treatment, the activity of SOD enzyme is the highest in the KO lines and lowest in the OE lines (Figure 6g), which is consistent with the activity trend observed in POD (Figure 6h). Overexpressing *SlBBX31* reduced the activity of the protective enzymes, which impeded the timely elimination of excessive ROS within the plants. Consequently, this led to more harm to the plants. On the contrary, the KO lines exhibited increased SOD and POD enzyme activity and then improved their tolerance to heat stress. These results demonstrate that altered redox status may contribute to the improved tolerance to heat stress in KO plants.

### 2.8. SlBBX31 Is Involved in the Heat-Stress Response Gene Regulatory Networks

Genes involved in the heat-stress response, specifically *HSF* and *HSP*, are instrumental in mediating plant adaptation to heat stress [10]. The expression levels of *HSP17.6* (*Solyc08g062437.1*), *HSP70-1* (*Solyc09g011030.3*), *HSP90* (*Solyc08g079170.3*), and *HSF24* (*Solyc02g090820.3*) peaked at 2 h of heat-stress treatment (Figure 7 and Appendix A). During this period, the KO lines showed much higher expression levels compared to the WT, while the OE lines exhibited notably lower levels. At 4 h of heat-stress treatment, although the transcription levels of these early heat-stress response genes were downregulated, their expression remained higher in the KO lines and lower in the OE lines. These gene expression patterns aligned with the phenotype in which OE lines were more sensitive to heat stress, while KO lines displayed greater heat tolerance. These findings speculate that *SlBBX31* may be involved in heat-stress response via the *HSF-HSP* signaling pathway.

Moreover, in this research it was discovered that the relevant genes encoding antioxidant enzymes, namely *POD* (*Solyc01g105070.3*), *SOD2* (*Solyc06g048410.4*), and *APX-2* (*Solyc09g007270.3*), also shared a similar expression pattern (Figure 7 and Appendix A). Both *POD* and *APX-2* reached their peak expression levels at 2 h of heat-stress treatment. During the process from 0 to 2 h, the KO lines manifested the greatest increase in expression, followed by the WT, while the OE lines displayed minimal changes in expression. As for the *SOD2* gene, all these lines exhibited an initial increase followed by a decrease in expression. From 2 to 12 h of treatment, the KO lines maintained higher expression levels than the WT, while the OE lines had lower expression. These data suggest that *SlBBX31* may participate in heat-stress response through the ROS scavenging pathway.

Finally, this study also examined the expression levels of genes related to hormone synthesis and signal transduction. *9-cis epoxycarotenoid dioxygenase* (*NCED*) is a key rate-limiting enzyme in plants which catalyzes the synthesis of ABA, and its catalytic process represents a critical step in the ABA biosynthesis pathway [10,11,18]. As the duration of heat treatment increased, the expression level of *NCED3* (*Solyc07g056570.1*) showed an overall upward trend (Figure 7 and Appendix A). Under varying treatment durations, there were slight differences in the expression of *NCED3* between the OE and WT lines. In contrast, the KO lines exhibited the highest expression levels which suggested that more ABA signaling molecules may be synthesized in the KO lines. The ABA signal transduction pathway is one of the most important pathways for plants to resist abiotic stress. The ABF transcription factor, a bZIP-type transcription factor which specifically recognizes ABA-responsive elements, contributes to the ABA signal transduction pathway [10,11,18]. We noted that during heat-stress treatment from 0 to 12 h, the expression of *ABF3* (*Solyc01g108080.4*) manifested an increasing trend in both KO and WT lines, while it decreased in the OE lines (Figure 7 and Appendix A). At different time points, the expression level of *ABF3* was higher in the KO lines, indicating that *SlBBX31* may affect tomato tolerance to heat stress through the ABF-mediated ABA signal transduction pathway. Aminocyclopropanecarboxylic acid synthase (ACS) and aminocyclopropanecarboxylic acid oxidase (ACO) are key enzymes regulating ethylene biosynthesis in plants [36]. In both KO and WT lines, the gene expressions of *ACO4* (*Solyc02g081190.4*) and *ACS6* (*Solyc08g081550.4*) were almost unaffected by heat stress (Figure 7 and Appendix A). Nevertheless, the OE lines showed much higher expression levels at various time points of heat-stress treatment compared to the WT and KO lines. We speculate that *SlBBX31* may alter ethylene content within the plant. ERF transcription factors are involved in plant defense against biotic and abiotic stresses by participating in signal transduction pathways for various hormones such as ETH [14], JA [17], and SA [15]. During heat-stress treatment at 0, 2, and 4 h, the expression level of *ERF5* (*Solyc03g093560.1*) was obviously higher in the OE lines compared to the KO and WT lines (Figure 7 and Appendix A). However, at 12 h of heat-stress treatment, the KO lines manifested notably higher *ERF5* expression than both the WT and OE lines. The elevated expression of *ERF5* in the KO strain might contribute to its enhanced heat tolerance. Lipoxygenase (LOX) is a key regulatory factor in the biosynthesis of JA and SA [37]. The expression level of *LOX1* (*Solyc08g029000.3*) was particularly low in both KO and WT lines, but much higher in the OE lines (Figure 7 and Appendix A), which may alter the contents of JA and SA in plants. In summary, *SlBBX31* may affect tomato tolerance to heat stress through hormone synthesis or hormone-mediated signal transduction pathways involving ABA, ETH, JA, and SA.

## 3. Discussion

Emerging evidence has demonstrated the crucial role played by BBX proteins in light-dependent pigment accumulation. Specifically, HY5, a key bZIP transcription factor, acts as a central coordinator which integrates light signals and environmental cues [38,39]. For instance, *AtBBX21* enhances both the activity and expression of AtHY5. As a result, it drives the expression of flavonoid biosynthesis genes and promotes anthocyanin accumulation [40]. Likewise, AtBBX22 interacts with and activates AtHY5, which leads to upregulated expression of flavonoid biosynthesis genes under AtHY5 control [41]. Interestingly, while overexpressing *SlBBX20* delays plant growth, it simultaneously promotes the buildup of chlorophyll, anthocyanins, and carotenoids in foliage [25]. Our phylogenetic analysis points out that SlBBX20 is closely related to AtBBX21 and AtBBX22. All the three genes are positive regulators of pigment accumulation. Although the functions of homologous proteins among different species are not necessarily conserved, this comparison also has certain reference value. Conversely, while *AtBBX31* negatively modulates anthocyanin accumulation by suppressing the activity of the AtBBX21–AtHY5 complex under white light, it serves as a positive regulator of UV-B signaling [42]. *AtBBX30* and *AtBBX31* are both negative regulators of light signaling [43]. In this study, *SlBBX31*-OE plants had largely reduced total chlorophyll content in the leaves, similar to the phenotype of *AtBBX31* under white light. The net photosynthetic rate of *SlBBX31*-OE plant leaves was notably decreased. Both Fv/Fm and Fv’/Fm’ values were much lowered, which indicated a decline in photosynthetic efficiency. The reduction in chlorophyll content disturbs the plants’ ability to convert light energy into chemical energy [25]. Therefore, the reduction in chlorophyll content and net photosynthetic efficiency caused by the *SlBBX31* overexpression might affect the accumulation of carbon and nitrogen sources necessary for plant growth. In view of the functional distinctions of *AtBBX31* under UV and white light conditions, the next step is to verify whether the function of *SlBBX31* also conforms to this rule under UV conditions. Given the paler fruits in *SlBBX31*-OE plants, subsequent steps could include measuring pigment content during different fruit development stages and analyzing transcriptome data to explore the impact of *SlBBX31* on pigment accumulation.

Antenna proteins are associated with both PS II and PS I, and all these protein complexes are closely related to photosynthesis [44]. Through further transcriptome sequencing analysis, it was discovered that while the expression levels of genes belonging to the LHC antenna protein family in *SlBBX31*-OE plants were apparently downregulated, the expression levels of LHC in *SlBBX31*-KO plants were notably higher. The photosynthetic apparatus in higher plants and algae is located on the thylakoid membrane, where PS II, PS I, cytochrome b6/f complex, and ATPase complex are present [44]. LHCs participate in capturing and conveying light energy, providing photoprotection and preserving the integrity of the thylakoid membrane. The PSII antenna proteins consist primarily of six pigment complexes: Lhcb1, Lhcb2, Lhcb3, Lhcb4, Lhcb5, and Lhcb6. On the other hand, the PSI antenna proteins are mainly comprised of four pigment protein complexes encoded by the genes *Lhca1*, *Lhca2*, *Lhca3*, and *Lhca4* [45,46]. The differentially expressed antenna protein genes in the OE and KO lines encompassed nearly all types of *LHCs* found in both PSII and PSI. The remarkable downregulation of *LHC* expression in the OE lines can lead to a decrease in antenna protein content, which will affect the photosynthetic response in tomato. Emerging study has identified the binding motifs of SlBBX31 from DAP-seq experiments [31]. Therefore, the following strategy is to verify whether SlBBX31 can directly regulate these *LHC* genes through subsequent dual luciferase reporter transactivation assays.

Certain BBX proteins are essential for the flowering pathway. AtBBX1 stands as a key regulatory factor in the photoperiodic regulation of flowering. It interacts with circadian clock genes and positively regulates downstream flowering genes. As a result, the expression of these genes is promoted [8]. AtBBX6 can influence the expression of the *FT* gene and thus affect the flowering time of *Arabidopsis* [47]. Under long daylight conditions, the OE lines of *AtBBX32* appear as a delayed flowering phenotype which illustrates that *AtBBX32* negatively regulates the flowering process [26]. Our study also found that *SlBBX31* has a negative effect on the flowering and fruiting stages. The *SlBBX31*-OE lines showed an evident decrease in pollen viability and a notable reduction in fruit setting rate. In addition, the fruiting nodes were higher, which indicated that overexpression of *SlBBX31* delayed the flowering and fruiting process. Meanwhile, the average fruit weight and diameter were decreased, leading to decreased crop yields. This study demonstrates that an excess of *SlBBX31* expression impacts the process of reproductive growth. Particularly, our work is not enough to prove how *BBX31* affects the flowering and fruiting processes. We hypothesize that it may affect photomorphogenesis through HY5 or CONSTITUTIVELY PHOTOMORPHOGENIC 1 (COP1) [39,40,41], which requires a considerable amount of follow-up research to verify this point in tomato.

BBX proteins can participate in stress responses to enhance plant tolerance to both biotic and abiotic stresses [48]. *AtBBX18* reduces plant heat tolerance by negatively regulating the expression of heat-responsive genes [49]. The apple BBX protein MdBBX37 positively regulates cold resistance in apples through the JAZ-BBX37-ICE1-CBF pathway [50]. A systematic identification and analysis of the tomato BBX family outlined that most tomato BBX proteins can be induced by heat, drought, and exogenous hormones, which suggests that tomato BBX proteins may be involved in the regulation of abiotic stress responses and hormone metabolism [51]. *SlBBX17* serves as a positive regulator in enhancing heat-stress tolerance [27]. SlHY5 is a factor in regulating ion absorption, metabolite buildup, and resilience against abiotic stresses in tomato [31,52,53,54]. SlHY5 has the ability to directly bind to the *SlBBX31* promoter and then it can activate *SlBBX31* transcription. Additionally, SlBBX31 can bind directly to the promoters of *SlCBF1/2* to activate their transcription, potentially serving as an important regulator in transcriptional activation of cold-induced *SlCBF*s, which leads to cold tolerance [31]. Our research filled the gap of the *SlBBX31* role under heat stress. Further transcriptome sequencing analysis revealed that, at the three time points of 2, 4, and 12 hafter heat-stress treatment, while the expression levels of *HSF* and *HSP* genes were lower in the *SlBBX31*-OE lines, they were obviously elevated in the *SlBBX31*-KO lines. Additionally, the expression levels of genes related to antioxidant protective enzymes, such as *POD*, *SOD2*, and *APX-2*, were higher in the KO lines, but lower in the OE lines. Combined with enzyme activity measurements, the results imply that *SlBBX31* may control heat-stress tolerance by crosstalk between the *HSF-HSP* and ROS pathways. Moreover, it can also affect the expression of key genes involved in the biosynthesis and signal transduction of ABA, ETH, JA, and SA. However, more evidence is needed at the molecular level to explain whether *SlBBX31* is involved in hormone synthesis or hormone-mediated signaling pathways. Therefore, subsequent experiments could concentrate on searching for the interaction and regulation proteins of SlBBX31 in order to elucidate the mechanism of its function under heat stress.

In conclusion, we focused on exploring the gene function of *SlBBX31* by using a reverse genetics approach. *SlBBX31*-OE plants displayed defects in photomorphogenesis and reproductive growth, along with an apparent decrease in heat-stress tolerance. On the contrary, *SlBBX31*-KO plants enhanced heat tolerance. Our present work illustrates that *SlBBX31* serves as a negative regulator for plant growth and heat-stress tolerance. This will provide useful information for the future design of a heat-tolerant tomato line.

## 4. Materials and Methods 

### 4.1. Plant Materials and Tomato Transformation

We obtained *S. lycopersicum* cv. M82 (M82) and *S. lycopersicum* cv. Ailsa Craig (AC) seeds from the Tomato Genetic Resource Center, University of California, Davis, CA, USA. 

For the construction of overexpressing vector, the full-length ORF of SlBBX31 was amplified by PCR using primers (OE-SlBBX31-F and OE-SlBBX31-R, sequences listed in Appendix A) and subsequently cloned into the transient expression vector pHELLSGATE2. 

For CRISPR-Cas9 knockout vector construction, the single-guide RNA (sgRNA) sequence targeting the *SlBBX31* gene was selected using the CRISPR-PLANT website (http://www.genome.arizona.edu/crispr/, accessed on 1 September 2020). SlBBX31-SgRNA-F and SlBBX31-SgRNA-R (Appendix A) were designed to construct the sgRNA-Cas9 vector, utilizing the vector backbone of pCAMBIA2301. 

Transgenic lines in AC background were achieved through tissue culture-based *Agrobacterium*-mediated genetic stable transformation (strain GV3101). 

### 4.2. Plant Growth Conditions and Stress Treatments

We sowed the seeds evenly in 7 × 7 cm pots with stroma (nutritive soil:vermiculite:perlite = 3:1:1) and placed them in artificial climate chambers. The optimal climate conditions were as follows: 16/8 h day/night photoperiod, 25 °C air temperature, 800 μmol/m^2^/s^1^ irradiance, and 70% humidity. We watered the plants each day to meet the optimal water requirement before the stress treatments.

One-month-old AC plants, *SlBBX31*-OE plants, and *SlBBX31*-KO plants of the same stage with maximum water holding capacity of the soil were subjected to 42 °C heat-stress conditions with 90% relative humidity. Samples, each consisting of pooled leaves from multiple individual plants with three biological replicates per line, were collected at 0, 2, 4, and 12 h.

Twenty-eight days after sowing, M82 plants of the same stage were subjected to 42 °C heat-stress conditions with 90% relative humidity for 0 h (Heat-0 h), 2 h (Heat-2 h), 4 h (Heat-4 h), 12 h (Heat-12 h), and 24 h (Heat-24 h), and returned to optimal conditions for 24 h (heat-recovery) after the heat-stress treatment. Leaves from the three biological replicates were immediately frozen in liquid nitrogen and stored at −80 °C for RNA sequencing.

### 4.3. RNA Extraction and qRT-PCR Analysis

The extraction of total RNA was carried out using the TRNzol Universal Total RNA Isolation Kit (sourced from Tiangen, Beijing, China), strictly adhering to the manufacturer’s guidelines. From this extracted RNA, 1 µg was utilized for the synthesis of first-strand cDNA, employing the HiScript II 1st Strand cDNA Synthesis Kit (supplied by Vazyme, Nanjing, China). qRT-PCR was performed as described previously [55]. Each gene was subjected to three biological replicates, with *SlActin7 (Solyc03g078400.3.1)* [20] serving as the internal reference gene. The relative expression levels of the genes were measured by applying the 2^−ΔΔCt^ method. For reference, the primers employed in this study are enumerated in Appendix A (named “qPCR” in the table).

### 4.4. Protein Physicochemical Properties Prediction and Phylogenetic Analysis 

The ExPASY website (http://web.expasy.org/, accessed on 3 September 2020) was utilized to acquire physicochemical data, including the isoelectric point, protein instability index, molecular weight, and hydrophilicity. For the construction of a phylogenetic tree, the neighbor-joining method in MEGA7 (http://www.ddooo.com/softdown/141094.htm, accessed on 7 September 2020) was employed. Additionally, the Evolview website (https://evolgenius.info//evolview-v2/, accessed on 7 September 2020) facilitated the visualization of the phylogenetic tree. 

### 4.5. RNA Sequencing and Data Analysis 

Next-generation sequencing was performed using Illumina HiSeq4000 at the Annoroad Technologies Corporation in Beijing, China. The RNA-seq data acquired were subsequently analyzed by the Trimmomatic (v0.39) data-trimming program [56] to trim and assess the quality of the raw paired-end reads. Trimmomatic, which can be downloaded from http://www.usadellab.org/cms/index.php?page=trimmomatic (accessed on 1 March 2020), was utilized with the following parameters: ILLUMINACLIP: adapter.fa:2:30:10 SLIDINGWINDOW:4:15 MINLEN:75. We used TopHat2 (v2.1.1) software [57] in orientation mode to individually align the clean reads with the reference genome (ITAG4.0, accessible at https://solgenomics.net/organism/Solanum_lycopersicum/genome/, accessed on 1 March 2020). TopHat2, which can be downloaded from http://ccb.jhu.edu/software/tophat/tutorial.shtml (accessed on 1 March 2020), was employed with the following parameters: -m 1 -g 1 -I 10000 --library-type=fr-firststrand. We used Cuffdiff, a program within Cufflinks (v2.2.1) [56,58], for differential expression analysis. Cufflinks was downloaded from http://cole-trapnell-lab.github.io/cufflinks/manual/ (accessed on 1 March 2020), and the parameters used were: --max-bundle-frags 1000000000 -I 10000 -u --library-type=fr-firststrand. The DEGs between two samples were selected using the following criteria: |log2FC(fold change)| ≥ 1 && fdr (false discovery rate probability) ≤ 0.05. We utilized DIAMOND (v2.0.15) [59] software for KEGG annotation, InterProScan (v5.34-73.0) [60] software for GO annotation, and finally, employed ClusterProfiler (v4.4.4) [61] to conduct enrichment analysis of GO and KEGG using the hypergeometric test method. Finally, we used the ggplot2 (v3.2.0) R package for visualization.

### 4.6. Subcellular Localization

The ORF of *SlBBX31*, excluding the stop codon, underwent amplification utilizing specific primers, namely SlBBX31-GFP-F and SlBBX31-GFP-R (Appendix A). Following amplification, the ORF was cloned into the transient expression vector pBI121-GFP. This GFP-fusion construct, designated as 35S-SlBBX31-GFP, was then transformed into *Agrobacterium* (strain GV3101). Following incubation of the agrobacterium to an OD_600_ of 0.6 at 28 °C, it was resuspended in an infiltration medium and introduced into tobacco leaves. The infiltrated leaves were initially cultured in darkness for one day, followed by an additional day under light conditions. To visualize nuclei, DAPI staining was co-infiltrated. Two days after infiltration, fluorescence signals from 35S-SlBBX31-GFP and 35S-GFP in the tobacco leaves were measured using the Olympus BX53 microscopy system (Olympus Microsystems, Tokyo, Japan).

### 4.7. Measurement of Chlorophyll Content and Photosynthetic Correlation Indices 

The third fully expanded true leaves from the apex of one-month-old AC plants, *SlBBX31*-OE plants, and *SlBBX31*-KO plants were carefully detached for the assessment of chlorophyll content and Pn. The chlorophyll content was tested by an established method [62]. A portable photosynthesis system (LI-6800; LI-COR, Lincoln, NE, USA) equipped with an infrared gas analyzer was used to measure the Pn. These measurements were conducted under controlled conditions: temperature of 25 °C, light intensity of 800 µmol·m^−2^·s^−1^, relative humidity of 70%, and a CO_2_ concentration of 400 µmol mol^−1^.

### 4.8. Measurement of Pollen Vigor and Yield-Related Indices 

Pollen viability was tested by using the TTC (2,3,5-triphenyltetrazolium chloride) staining method [63]. 

Five tomato lines, namely AC, OE-2, OE-5, KO-11, and KO-20, were planted in 2019, with 15 plants for each line. In 2020, 10 plants were planted for each of the five strains. Data such as the number, weight, diameter, seed count within the fruit, and fruiting node of each tomato harvested were collected. Two fruiting branches at the same fruiting node were selected and marked on each tomato plant, and the fruit setting rate was calculated based on the total number of flowers and the actual number of fruits on these two branches.

### 4.9. Analysis of the H_2_O_2_ and O^2•–^

To detect the accumulation of H_2_O_2_ and O^2•−^, 3,3′-diaminobenzidine (DAB) and nitro blue tetrazolium (NBT) staining were employed. Specifically, leaves from the same position on both control and transgenic plants were immersed in DAB (1 mg·mL^−1^) or NBT (0.5 mg·mL^−1^) solutions for an entire night under dark conditions. After achieving adequate bleaching by boiling the leaves in 75% (v/v) ethanol, photographs were taken. For the quantitation of H_2_O_2_, the approach outlined by Xia et al. [64] was adopted, and the test of O^2•−^ levels relied on the method put forth by Liang et al. [55].

### 4.10. Assessment of Antioxidant Enzyme Activities

Leaf samples (weighing 100 mg each) were pulverized in 1 mL of 100 mM phosphate buffer (pH 7.0), which had been pre-cooled with ice and fortified with 1 mM EDTA, 0.1% (v/v) Triton X-100, and 1% (w/v) PVP40. Following centrifugation of the homogenates at 13,000× *g* for 20 min at 4 °C, the supernatants were collected. These supernatants were then used to assess the activities of the antioxidant enzymes, SOD and POD, following previously established methodologies [20].

### 4.11. Data Statistics and Analysis

We used GraphPad Prism (v8.0.2) and SPSS (v15.0) software for data statistical analysis and plotting.

## Figures and Tables

**Figure 1 ijms-25-09289-f001:**
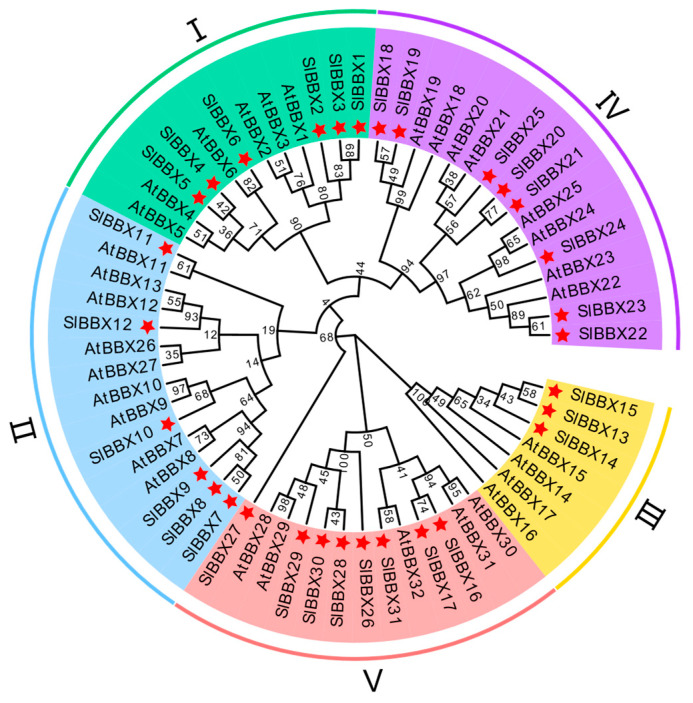
Phylogenetic relationships of BBX gene family members from tomato and *Arabidopsis*. The neighbor-joining method, with 1000 replicates, was employed to conduct phylogenetic analysis. To facilitate identification, the branches representing various subfamilies were labeled using distinct colors. The Roman numerals I–V represented the subfamily numbers respectively. Tomato BBX gene family members were marked with pentagrams.

**Figure 2 ijms-25-09289-f002:**
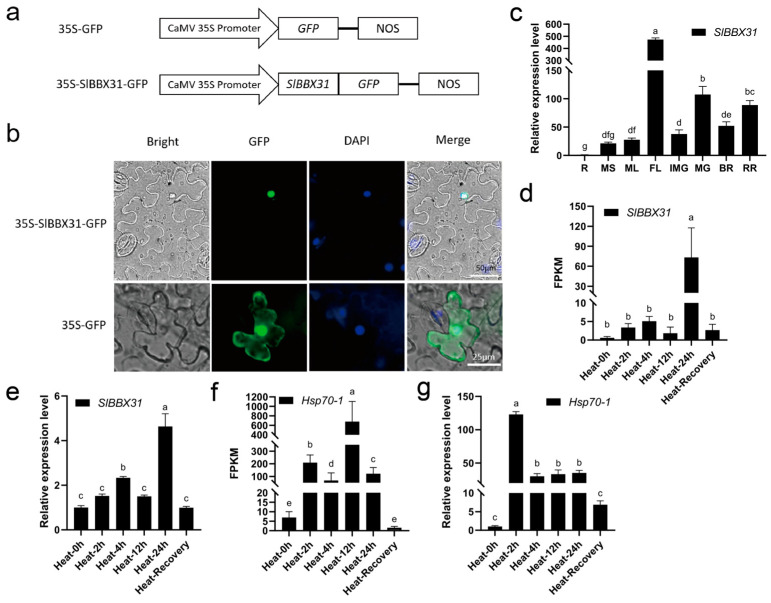
Expression patterns of *SlBBX31*. (**a**) Schematic diagram of the construction of the subcellular localization vector of *SlBBX31*. (**b**) The plant expression vectors, consisting of green fluorescent protein (GFP) fused with SlBBX31 proteins or the control vector, were introduced into tobacco leaves. Nuclei were visualized through DAPI staining. The term “Merge” refers to the combination of GFP and DAPI staining images. Scale bars, 50 µm and 25 µm, respectively. (**c**) Relative expression level of *SlBBX31* in different tissues of tomato. R: root, MS: Mature stem, ML: Mature leaf, FL: Flower, IMG: Immature green fruit, MG: Mature green fruit, BR: Breaker ripe stage fruit, RR: Red ripe stage fruit. (**d**) Expression analysis of *SlBBX31* in heat-stress transcriptome sequencing. (**e**) Relative expression analysis of *SlBBX31* in heat-stress condition. (**f**) Expression analysis of *Hsp70-1* in heat-stress transcriptome sequencing. (**g**) Relative expression analysis of *Hsp70-1* in heat-stress condition. *SlActin7* was selected as internal control. The relative gene expression level in other tissues was calculated by comparing the gene expression in roots. The least significant difference (LSD) was used for the significance test (*p*-value < 0.05). The different letters represent different levels of significance.

**Figure 3 ijms-25-09289-f003:**
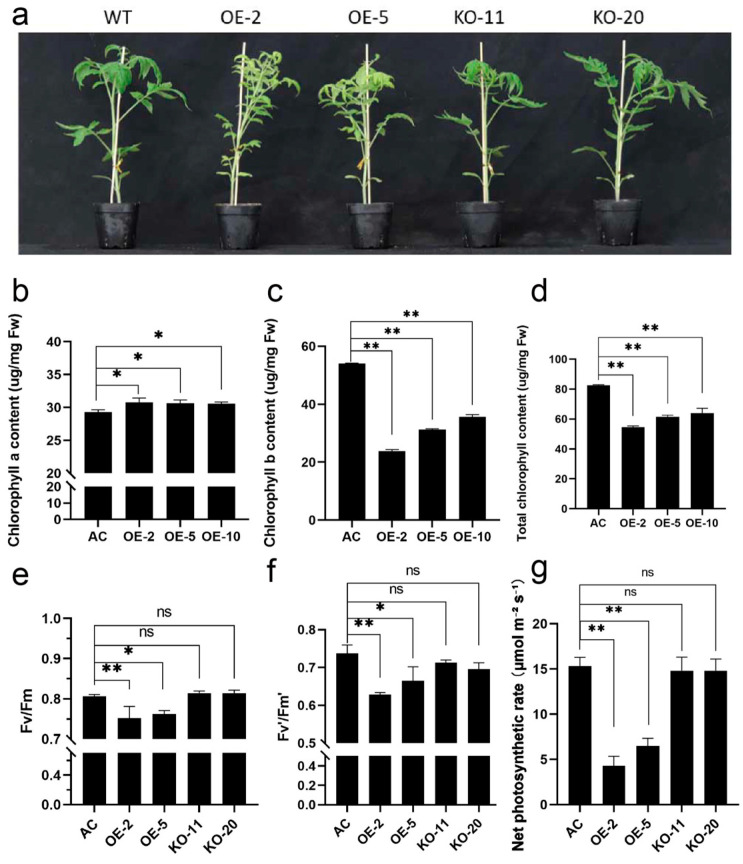
Overexpression of *SlBBX31* affected the vegetative growth of tomato. (**a**) Phenotype of transgenic plants. (**b**) Chlorophyll a content. (**c**) Chlorophyll b content. (**d**) Total chlorophyll content. (**e**) Fv/Fm value. (**f**) Fv’/Fm’ value. (**g**) Net photosynthetic rate. The asterisks represented significant differences compared with control (* represents *p*-value < 0.05, ** represents *p*-value < 0.01 and ns represents not significant).

**Figure 4 ijms-25-09289-f004:**
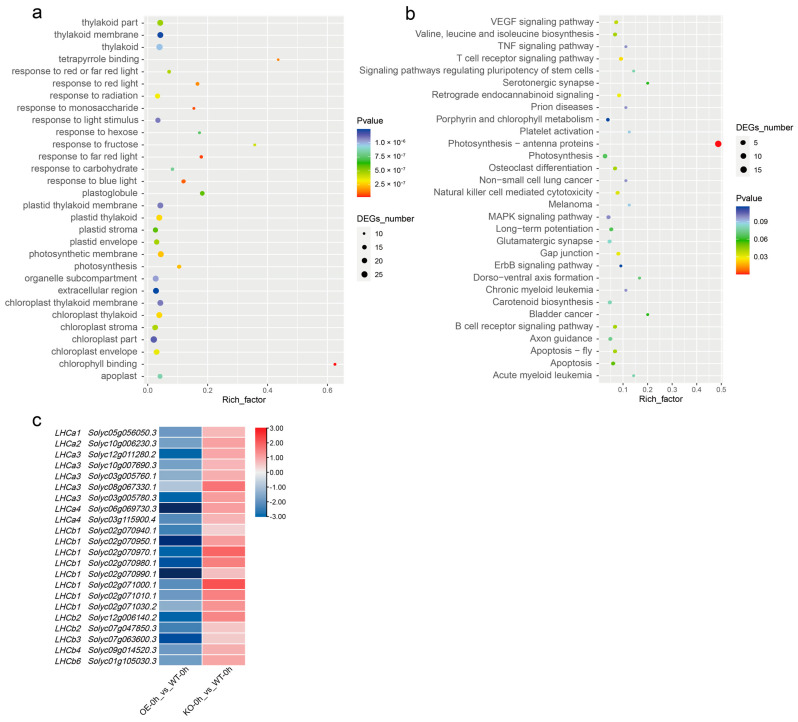
DEGs in OE and KO lines under normal growth conditions. (**a**) GO enrichment analysis of DEGs was performed. (**b**) KEGG enrichment analysis of DEGs was also conducted. (**c**) The differential expression patterns of antenna protein family genes in OE and KO lines were further investigated. The log2FC value for each gene was calculated based on its FPKM value.

**Figure 5 ijms-25-09289-f005:**
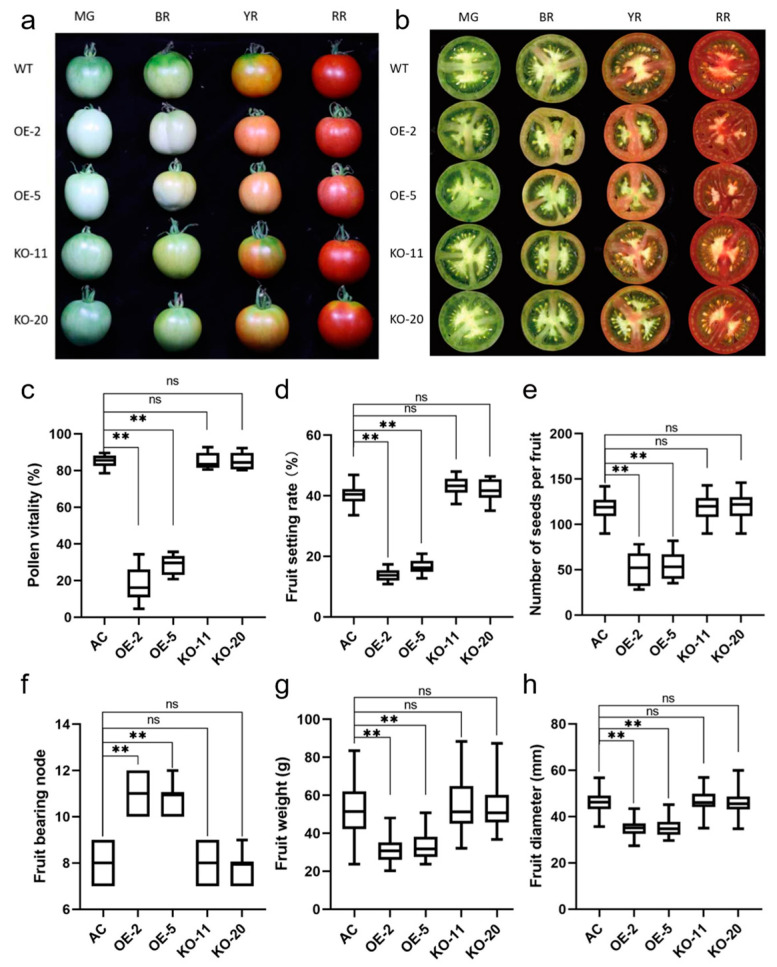
Overexpression of *SlBBX31* affected the reproductive growth of tomato. (**a**) Phenotype of the fruits of transgenic tomato. (**b**) Cross-section of the fruits of transgenic tomato. (**c**) Pollen viability. (**d**) Percentage of fertile fruit. (**e**) Number of seeds per fruit. (**f**) Fruit bearing node of tomato. (**g**) Fruit weight. (**h**) Fruit diameter. The asterisks represented significant differences compared with control (** represents *p*-value < 0.01 and ns represents not significant).

**Figure 6 ijms-25-09289-f006:**
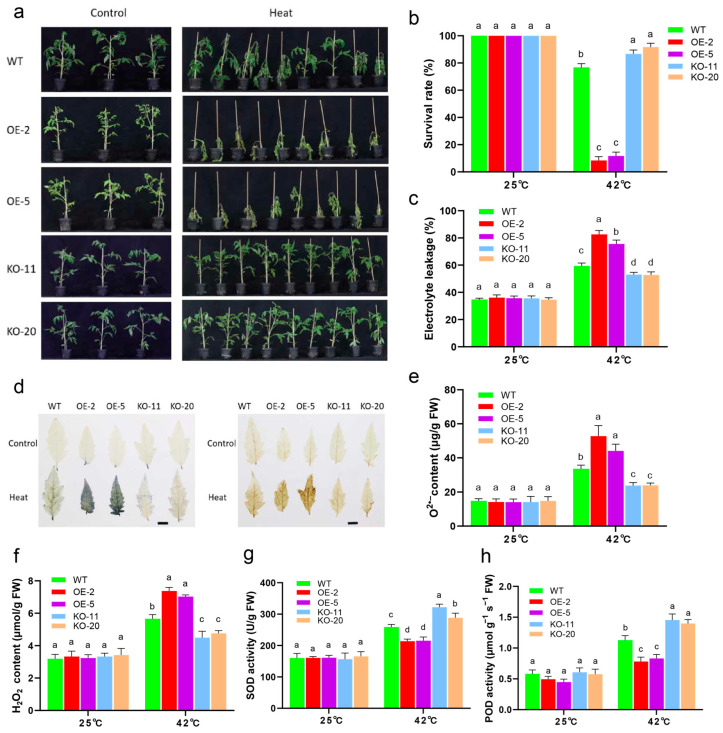
Functional analysis of *SlBBX31* in heat-stress response. (**a**) Phenotype of transgenic lines under heat-stress treatment. (**b**) Statistics of survival rate of transgenic lines under heat-stress treatment. (**c**) Electrolyte leakage rate of transgenic lines under heat-stress treatment. (**d**) NBT staining and DAB staining. (**e**) Determination of O^2•−^ content. (**f**) Determination of H_2_O_2_ content. (**g**) Determination of SOD activity. (**h**) Determination of POD activity. Statistical significance was determined using LSD test, with a *p*-value threshold of <0.05. The different letters represent different levels of significance.

**Figure 7 ijms-25-09289-f007:**
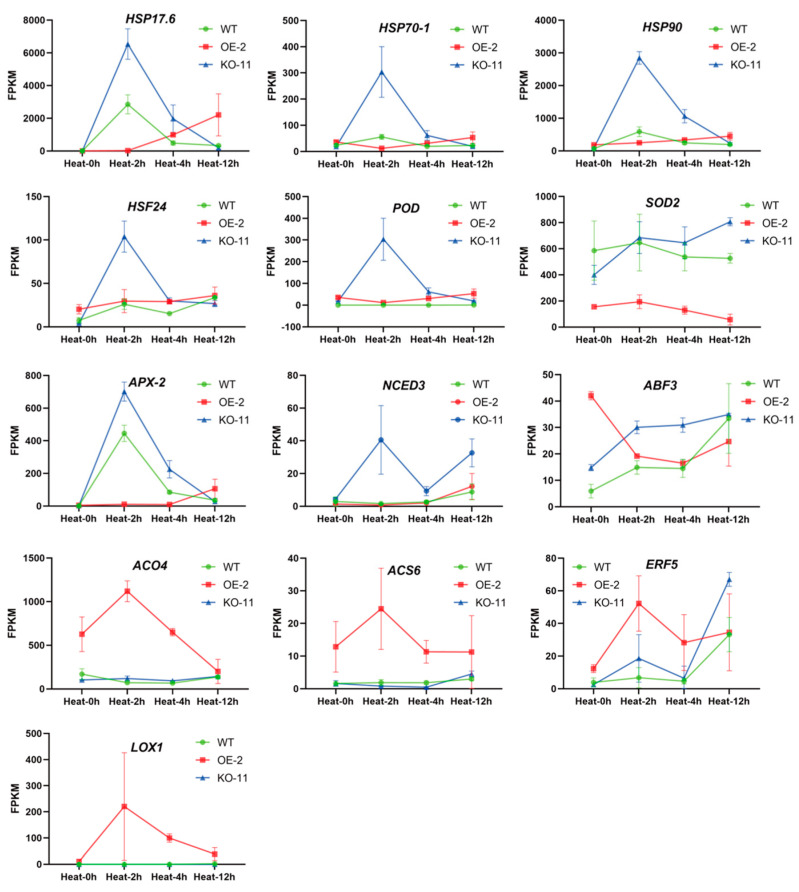
Analysis of key gene expression patterns following transcriptome sequencing conducted under heat-stress condition.

## Data Availability

We submitted the RNA-seq data at the NCBI Sequence Read Archive (SRA) under accession SRP262801. The gene expression data were uploaded to the NCBI Gene Expression Omnibus under accession number GSE151277. Other data supporting the conclusions of this study can be obtained from the corresponding author upon reasonable request. Please contact the authors for further information.

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
