# Peer review of "Elucidating the Role of SlBBX31 in Plant Growth and Heat-Stress Resistance in Tomato"

_ijms, 2024, doi:10.3390/ijms25179289_

Round 1

Reviewer 1 Report

Comments and Suggestions for Authors

The manuscript entitled "Elucidating the role of SlBBX31 in plant growth and heat stress resistance in tomato" targets an important aim in Solanum lycopersicum research. However, I have some issues/comments that need the authors attention:

- The manuscript is overall well written but there are many sentences that do not match and rather seem to be written by some AI software. For instance, the introduction is well written but failing to introduce this study. The authors seem to write the introduction as if this was a review paper: "This review delves into the multifaceted impacts of heat stress 35 on plants, highlighting not only the direct physiological consequences but also the intricate 36 molecular mechanisms involved[3, 5].". Yet, this is a research paper. There are also multiple sentences in the introduction, abstract and discussion that seem odd in the context of this study.

- There is no clear definition of the aims of this research. The introduction should specify some hypothesis or at least the aims of this study.

- Methods are very hard to follow. The authors have picked two varieties, M82 and AC for this research, that were sown in growth chambers (which ones?). Then M82 plants were subjected to heat stress " 0 h (Heat-0 h), 2 h (Heat-2 h), 4 h (Heat-4 h), 12 h (Heat-12 h), and 24 h (Heat-24 521 h), and returned to optimal conditions for 24 h (Heat-Recovery). Which plants? How old?

-Interestingly, the authors also "We selected 28-day-old M82 seedlings of the same stage for the drought treatment". Despite the fact that authors have written and aimed for heat stress, they also examined the effects of drought. ?? Without further explanation this is not comprehensive. 

- The authors also subjected M82 variety to cold stress: "Twenty-eight days post-sowing, M82 plants at the same developmental stage, with 535 saturated soil, underwent cold stress conditions for various durations of 4°C and 90% relative 536 humidity: 0 h (Cold-0 h), 3 h (Cold-3 h), 6 h (Cold-6 h), and 12 h (Cold-12 h)." My comment here is the same as above.

- What have the authors done with the other variety ? It is stated in other experiments but not in the stress one.

- The methods used allow no replication of this study as the authors did not specified any of the parameters used in the software stated in the text.

- What was the reference genome used and how?

- " qRT-PCR was subsequently executed in accordance with the protocols outlined by the Bio-Rad CFX system (Bio-Rad, USA)." - this is another example of a vague description of what was performed.

- How many genes were used in rt-PCRs and why? I know they are stated in Table S1 but it is hard to understand some (pCAMBIA2301-F?, 35S-F?). Also, what is the reference of these genes?

- What was the correlation with transcriptome results?

Figures: the text of several figures is impossible to read.

Proofreading: please italicize all genes. The species name should also be written with the author of the species the first time it is written.

Comments on the Quality of English Language

See above.

Reviewer 2 Report

Comments and Suggestions for Authors

The manuscript by Wang and Zhan presents a solid effort in studying the role of the SlBBX31 protein in tomato growth and heat stress resistance. The research is thorough and provides important information on the molecular mechanisms underlying these processes, and also contributes to the understanding of plant heat stress responses. However, there are many areas that could benefit from further refinement and attention to detail:

1.   The manuscript appears to have been generated with great input from AI tools, as evidenced by repetitive language and phrasing. For instance, the authors should take care about the use of word “significantly” only when referring to results supported by statistical analysis, but not to describe general observations/effects. In the abstract alone, the word appears three times in different contexts. In the manuscript are also frequently mentioned certain words used by ChatGPT in scientific writing, such as “crucial” (13 times), “pivotal” (6 times) and “thereby” (12 times), and also unnecessary apostrophes for the possessive form “lines 228, 281, 351, 421, etc.”. I recommend greater human involvement to improve the precision and avoid over-reliance on specific terms.

2.   A sentence in the abstract that states "This suggested that the overexpression of SlBBX31 significantly impacted the growth process of tomato, specifically in terms of increased growth rate and biomass production", but appears to be incorrect based on the findings presented in the manuscript. This should be revised to reflect the results.

2. The key words are too general and do not adequately capture the unique aspects of the study. More specific keywords should be included to enhance the manuscript visibility.

3.   Lines 80-87: The introduction end summarises the results but does not clearly state the main research hypothesis. This part should be modified to present the research question/hypothesis and a clear focus for the study.

4.   In Figure 2b, DAPI stains three nuclei, but GFP is detected in only one. What could be possible reasons for that - variations in transfection efficiency, differences in SlBBX31-GFP expression among the cells or technical issues with imaging or protein localization? Clarification is needed to ensure accurate interpretation of the results.

5.   The authors claim that overexpression of SlBBX31 slows plant development, but the manuscript lacks quantitative data to support this statement. Images in Figs 4a & 7a are illustrative but do not provide the required quantification of phenotypic changes under normal and heat stress conditions, and how the overall plant growth and development are affected by the overexpression or downregulation of SlBBX31. Metrics such as plant height, biomass or growth rate should be included to support these claims. It is also a bit strange that under control conditions all lines show similar survival rates.  

6.   Lines 422-438: The function of SlBBX31 does not appear to be conserved across different species, as shown by contrasting effects in tomato versus Arabidopsis. The authors should discuss this contrasting role and potential reasons and implications for BBX protein functions across species.

7.   Lines 423-426: contain a repetitive statement: “overexpressed AtBBX31 boosts the accumulation of chlorophyll and anthocyanins in plants, possibly via an AtHY5-dependent pathway. Furthermore, under these conditions, overexpressing AtBBX31 also enhances the accumulation of chlorophyll and anthocyanin content[40]” This redundancy should be eliminated.

8.   In general, the Discussion section would benefit from a more in-depth and critical analysis of the obtained results.

9.   Is the irradiance level of 800 μmol/m²/s appropriate for tomato plants? This intensity seems quite high and could potentially cause photooxidative stress and leaf burn or reduced photosynthetic efficiency.

10. Some experimental methods are only briefly mentioned or not described at all. For example, the manuscript lacks information on CRISPR vector construction, the generation of OE lines, tomato transformation protocols, the online tools used for sequence and phylogenetic analysis, GO analysis, statistical methods, etc. I recommend to revise the MM section and provide thorough descriptions of all techniques used to obtain the results.

Minors:

11.  Line 10: The abbreviation “BBX” should be used to refer to “B-box” and not to “B-box zinc-finger proteins”.

12.  Ensure that there is a space between the last word of a sentence/phrase and the citation brackets, this spacing is missing throughout the text and needs to be corrected.

13.  Line 179: In the manuscript are occasionally used contractions like “It’s worth noting”, which can undermine formality. Avoid contractions to maintain a formal tone throughout the manuscript.

Comments on the Quality of English Language

The English is fine, but the manuscript contains repetitive language and phrasing, which could be streamlined for better readability.

Round 2

Reviewer 1 Report

Comments and Suggestions for Authors

I would like to thank authors for the explanations provided and how changes were performed in this new version. The explanation made by the authors allowed me to understand better what was done. However, I still have one minor comment. Since the reviewers have analyzed multiple stresses it is not easy to understand the focus on heat stress. This starts by the tile "Elucidating the role of SlBBX31 in plant growth 3 and heat stress resistance in tomato" and continues in the introduction and aims. Yet, the authors have studied other factors that simply appear in results without further explanations. 

Comments on the Quality of English Language

See above.

Author Response

Comments 1: I would like to thank authors for the explanations provided and how changes were performed in this new version. The explanation made by the authors allowed me to understand better what was done. However, I still have one minor comment. Since the reviewers have analyzed multiple stresses it is not easy to understand the focus on heat stress. This starts by the tile "Elucidating the role of SlBBX31 in plant growth 3 and heat stress resistance in tomato" and continues in the introduction and aims. Yet, the authors have studied other factors that simply appear in results without further explanations. 

Response 1: Thank you for pointing this out. I agree with your comment. Therefore, we have deleted all data pertaining to drought and chilling stress, and solely focused on discussing the results of heat stress in the paper. Kindly refer to the second revised version of the paper for further details.

Reviewer 2 Report

Comments and Suggestions for Authors

The authors have generally agreed with the comments provided and made revisions accordingly. However, some comments have not been fully addressed.

Comment & Response 5The comment raised a concern about the discrepancy between DAPI-stained nuclei and GFP-detected ones, which the authors did not directly address. They stated that "the signals from the SlBBX31-GFP fusion protein overlapped with those from the DAPI staining", and recognise the issue of low transfection efficiency in their response to the comment, but did not provide a clear explanation for this discrepancy. Despite this, their response could still be considered acceptable, as the observed variability does not undermine the conclusion regarding SlBBX31-GFP nuclear localisation.

Comment & Response 6: Although the authors agreed with the comments, they did not provide quantitative data to support the claims regarding plant growth. To strengthen their argument, they should provide sufficient quantitative evidence to support their claims about how the overexpression or downregulation of SlBBX31 affects overall plant growth and development.

Comments on the Quality of English Language

The manuscript has grammatical errors and awkward phrasings that require polishing. A thorough proofreading to correct these issues is needed.

Author Response

Comment & Response 5: The comment raised a concern about the discrepancy between DAPI-stained nuclei and GFP-detected ones, which the authors did not directly address. They stated that "the signals from the SlBBX31-GFP fusion protein overlapped with those from the DAPI staining", and recognise the issue of low transfection efficiency in their response to the comment, but did not provide a clear explanation for this discrepancy. Despite this, their response could still be considered acceptable, as the observed variability does not undermine the conclusion regarding SlBBX31-GFP nuclear localisation.

Response 5: Thank you for pointing this out. I agree with your comment. I apologize for any confusion caused. I agree with your comment. We have explained this issue in Section 2.2 of the article (Lines 110-116).

Comment & Response 6: Although the authors agreed with the comments, they did not provide quantitative data to support the claims regarding plant growth. To strengthen their argument, they should provide sufficient quantitative evidence to support their claims about how the overexpression or downregulation of SlBBX31 affects overall plant growth and development.

Response 6: The growth process of plants includes vegetative growth and reproductive growth. Previously, our conclusion was not fully considered, and our data was not sufficient to support that BBX affects the vegetative growth process in tomato. Therefore, we have changed our conclusion to that SlBBX31 affects chlorophyll accumulation and photosynthetic processes (Line 207). During reproductive growth, SlBBX31 affects seed development, flowering, and fruiting. Therefore, based on these two aspects, we have revised sections 2.4 and 2.5 of the paper (Lines 147-224).

Comments on the Quality of English Language: The manuscript has grammatical errors and awkward phrasings that require polishing. A thorough proofreading to correct these issues is needed.

Response: We have conducted a second revision of the article, and in order to facilitate your intuitive understanding of the specific modifications, we have highlighted all changes in green color. Kindly refer to the second revised version of the paper for further details.